



# Using specularity content to evaluate five geothermal heat flux maps of Totten Glacier

Yan Huang[1],Liyun Zhao[1,2*], Yiliang Ma[1], Michael Wolovick[3], John C. Moore[1,4*]
[1] College of Global Change and Earth System Science, Beijing Normal University,
Beijing 100875, China
[2] State Key Laboratory of Earth Surface Processes and Resource Ecology, Beijing
Normal University, Beijing 100875, China
[3] Glaciology Section, Alfred-Wegener-Institut Helmholtz-Zentrum für Polar- und
Meeresforschung, Bremerhaven, Germany
[4] Arctic Centre, University of Lapland, Rovaniemi, Finland
* Corresponding author
Corresponding author: Liyun Zhao (zhaoliyun@bnu.edu.cn);  John C. Moore
(john.moore.bnu@gmail.com)

## Abstract

Geothermal heat flux (GHF) is an important factor affecting the basal thermal
environment of an ice sheet and crucial for its dynamics. But it is notoriously poorly
defined for the Antarctic ice sheet. We compare basal thermal state of the Totten Glacier
catchment as simulated by five different GHF datasets. We use a basal energy and water
flow model coupled with a 3D full-Stokes ice dynamics model to estimate the basal
temperature, basal friction heat and basal melting rate. In addition to the location of
subglacial lakes, we use specularity content of the airborne radar returns as a two-sided
constraint to discriminate between local wet or dry basal conditions and compare them
with the basal state simulations with different GHF. Two medium magnitude GHF
distribution maps derived from seismic modelling rank best at simulating both cold and
warm bed regions well, the GHFs from Shen et al. (2020), and from Shapiro and
Ritzwoller (2004). The best-fit simulated result shows that most of the inland bed area
is frozen. Only the central inland subglacial canyon, co-located with high specularity
content, reaches pressure-melting point consistently in all the five GHFs. Modelled
basal melting rates there are generally 0-5 mm yr$^{-1}$ but with local maxima of 10 mm yr$^{-1}$
. The fast-flowing grounded glaciers close to Totten ice shelf are lubricating their bases
with melt water at rates of 10-400 mm yr$^{-1}$.

## 1 Introduction

Totten Glacier is the primary outlet glacier of the Aurora Subglacial Basin (ASB), and
one of the most vulnerable glaciers to a warming climate in East Antarctica (Dow et al.,
2020). It holds an ice volume equivalent to 3.9 meters of global sea level (Morlighem
et al., 2020; Greenbaum et al., 2015). Most of the bedrock below Totten Glacier is
below sea level. Totten Ice Shelf has a relatively high basal melt rate of ~10 m yr$^{-1}$
compared with other ice shelves in East Antarctica (Rignot et al., 2013, Roberts et al.,
2018) and has thinned and lost mass rapidly in recent years (Pritchard et al., 2009;
Adusumilli et al., 2020).




The ASB has a widespread distributed hydrological network with almost 200 'lake-like'
or water accumulation features. There may be a hydrological flow pathway operating
from subglacial lakes near the Dome C ice divide and the coast via the Totten Glacier
(Wright et al., 2012), potentially affecting the stability of the Totten Glacier.

Basal melting may contribute to subglacial hydrological flow. Basal meltwater
lubricates the flow of ice, which can impact the stability of the ice sheet and the
direction of the ice flow (Livingstone et al., 2016; Bell et al., 2007). The basal meltwater
moves down the pressure gradient and gradually develops into a complex subglacial
hydrological system, which eventually flows into the ocean (Fricker et al., 2016).
However, the spatial structure of the basal thermal state and basal melting rates beneath
the Totten Glacier are not yet well understood.

Basal melting can occur where the ice temperature reaches the pressure melting point,
dramatically lowering the basal friction and allowing the ice to flow faster. Geothermal
heat flux (GHF) is an important boundary condition for ice temperature. Its magnitude
and distribution affect the distribution of basal ice temperature and thus ice flow. The
magnitude of GHF depends on the spatially varying geological conditions that control
heat generation and conduction, including heat flux from the mantle, crustal thickness,
heat production in the crust by radioactive decay, groundwater flow, and tectonic history
(Pollack et al., 1993; Pittard et al., 2016). It is difficult to measure GHF directly due to
limited access to Antarctic bedrock, with only a few point measurements in ice-free
areas or from boreholes through the ice (Fisher et al., 2015). GHF datasets are
commonly estimated from models relying on either seismic (Shapiro and Ritzwoller,
2004; An et al., 2015; Shen et al., 2020), airborne magnetic data (Martos et al., 2017),
or satellite geomagnetic data (Fox-Maule et al., 2005; Purucker et al., 2013).

Previous thermomechanical simulations of Totten Glacier (Dow et al., 2020; Pattyn et
al., 2010; Pittard et al., 2016; Van Liefferringe et al., 2018) have used GHF data from
Shapiro and Ritzwoller (2004), Purucker et al. (2013) and An et al. (2015), but Wright
et al. (2012) used spatially uniform values. In this study, we simulated the basal thermal
state of Totten Glacier, based on the best available topographic data and five different
GHFs, including three GHF listed above, plus more recent GHF fields from Martos et
al. (2017) and Shen al et. (2020).

We apply an off-line coupling between a basal energy and water flow model and a 3D
full-Stokes ice flow model for each of the 5 GHF maps, to provide the best-fit
distribution of modelled basal temperature and basal melt rate. We evaluate the
simulated basal temperature fields under the different GHF maps using the observations
of water at the ice base to infer which GHF map is most reliable in the ASB. The
observations include a set of subglacial lakes locations and the specularity content (Dow
et al., 2020) calculated from airborne radar data collected by the International
Collaborative Exploration of the Cryosphere by Airborne Profiling (ICECAP) survey.



Specularity is a parameterization of the along-track radar bed reflection scattering
function that has been used to provide an attenuation-independent proxy for distributed
subglacial water bodies (Schroeder et al., 2013). We devise measures of specularity that
help discriminate between alternative GHF maps to best characterize both cold and
warm beds.
**2 Regional Domain and Datasets**
Our modeled domain, the Totten Glacier, is located in the Aurora Subglacial Basin in
East Antarctica (Fig. 1). Its boundary is based on drainage-basin boundaries defined
from satellite ice sheet surface elevation and velocities (Mouginot et al., 2017). The
surface elevation, bedrock elevation, and ice thickness are from MEaSUREs
BedMachine Antarctica, version 2 with a resolution of 500 m (Morlighem et al., 2020).
Simulation input and comparison datasets are shown in Table 1. The surface ice velocity
data are obtained from MEaSUREs Phase-Based Antarctica Ice Velocity Map, Version
2 with resolution of 450 m (Rignot et al., 2017), which were mainly collected during
the International Polar Years from 2007 to 2009 with additional surveys between 2013
and 2016. Ice sheet surface temperature is prescribed by ALBMAP v1 with a resolution
of 5 km (Le Brocq et al., 2010) and comes from monthly estimates inferred from
AVHRR data averaged over 1982-2004 (Comiso, 2000). Subglacial lake locations are
from the fourth inventory of Antarctic subglacial lakes (Wright and Siegert, 2012) and
the first global inventory of subglacial lakes (Livingstone et al., 2022).
Five GHF datasets (Fig. 2; Table 2) are used in this study. All the datasets are
interpolated into 2.0 km resolution. The specularity content data are from Dow et al
(2020), where they calculated radar specularity content over ASB from the ICECAP
survey lines, and smoothed the data with a 1 km filter, following the equations described
in Schroeder et al. (2015). Specularity content is given as a relative value between 0
and 1, larger values mean a higher likelihood of the presence of water, and value of 0.4
is taken as the division where specularity content shows the presence of water (Young
et al., 2016).
Table 1 Datasets used in simulations.

| Variable name | Dataset | Resolution | Reference |
|---|---|---|---|
| surface elevation, bedrock elevation, and ice thickness | MEaSUREs BedMachine Antarctica version 2 | 500 m | Morlighem et al., 2020; Cui et al., 2020 |
| surface ice velocity | MEaSUREs InSAR-based Antarctic ice velocity Map, version 2 | 450 m | Rignot et al., 2017 |
| surface temperature | ALBMAP v1 | 5 km | Le Brocq et al., 2010 |
| subglacial lake location | The first global inventory of subglacial lakes | ----- | Livingstone et al., 2022 |
| specularity content | Aurora Subglacial Basin GlaDs inputs, outputs and geophysical data | 1 km along track | Dow et al., 2019 |






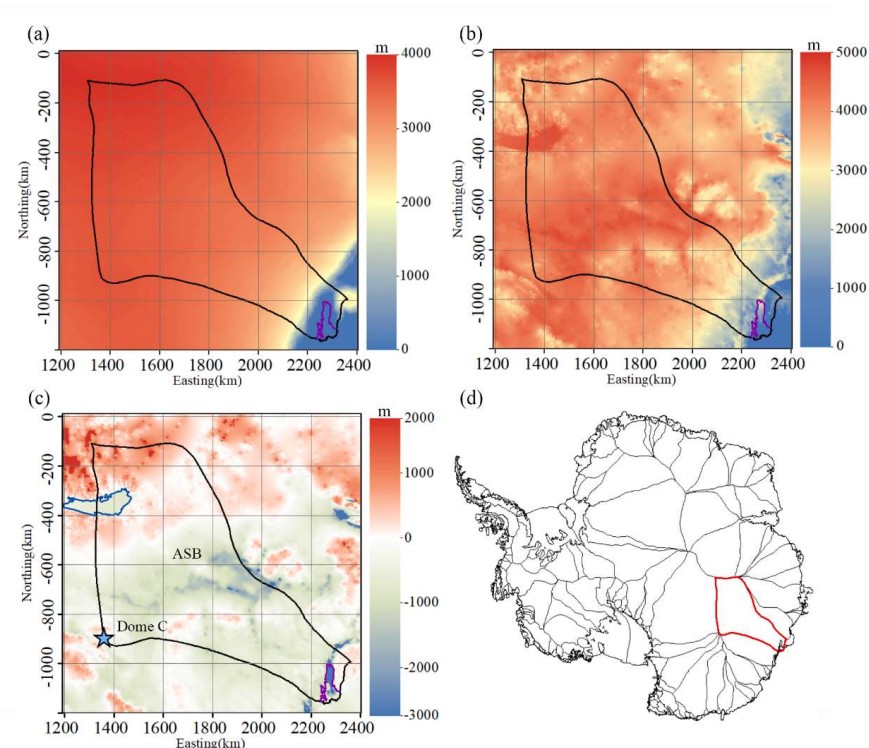


Fig. 1. The domain topography and location with domain boundary overlain. (a) surface elevation;
(b) ice thickness; (c) bed elevation; (d) the location of our domain in Antarctica. The solid black
curve is the outline of the study domain, including the Totten ice shelf. The purple curve in (a-c) is
the grounding line of Totten glacier. The blue curve in (c) is Lake Vostok (Studinger et al., 2003).
The solid red curve in (d) is the boundary of Totton Glacier. ASB and Dome C (blue star) are marked
in (c).

Table 2 The five GHF datasets used with the mean and range in our region.

| GHF map | Reference | Method | Mean (mW m$^{-2}$) | Range (mW m$^{-2}$) |
|---|---|---|---|---|
| Martos | Martos et al., 2017 | airborne geomagnetic data | 65 | 51-70 |
| Shen | Shen et al., 2020 | seismic model | 58 | 42-63 |
| An | An et al., 2015 | seismic model | 51 | 34-56 |
| Shapiro | Shapiro and Ritzwoller, 2004 | seismic model | 58 | 44-63 |
| Purucker | Purucker, 2013 | Satellite geomagnetic data | 51 | 37-67 |

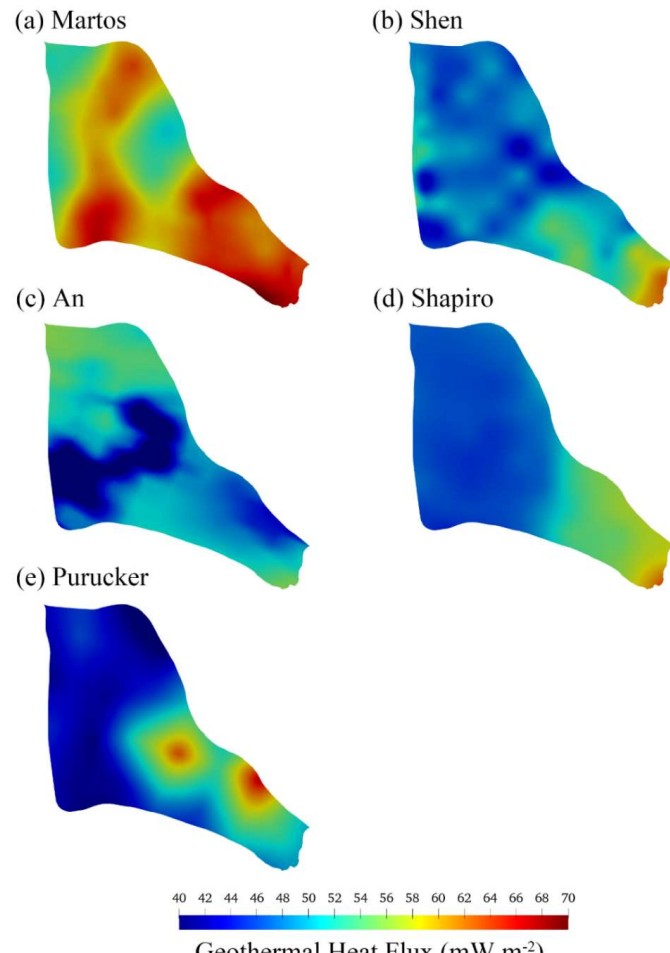


Fig. 2. The spatial distribution of GHF over our domain as described in Fig. 1. See Table 2 for the
GHF map details.

**3 Model**
Our goal is to map the basal thermal state of Totten glacier, including basal temperature
and basal melting rate. GHF, basal frictional heat and englacial heat conduction are the
main factors that determine the basal thermal state of the ice sheet. We need to simulate
the ice flow velocity and stress to calculate the basal frictional heat, and to simulate the
ice temperature to calculate the englacial heat conduction flux.

Following the same method as Kang et al. (2022), we solve an inverse problem by a
full-Stokes model, implemented in Elmer/Ice, to infer the basal friction coefficient such
that the modelled velocity best fits observations. To get a proper vertical ice temperature
profile subject to thermal boundary conditions needed in solving the inverse problem,




we use a forward model that consists of an improved Shallow Ice Approximation (SIA)
thermomechanical model with a subglacial hydrology model (Wolovick et al., 2021a).
We do steady state simulations by coupling the forward and inverse models.
**3.1 Mesh Generation and Refinement**
We use GMSH (Geuzaine and Remacle, 2009) to generate an initial 2-D horizontal
footprint mesh. Then we refine the mesh by an anisotropic mesh adaptation code in the
Mmg library (http://www.mmgtools.org/). The resulting mesh is shown in Fig. 3 and
has minimum and maximum element sizes of about 800 m and 20 km. The range of
mesh size is 800 m at ice shelf, 1-3 km upstream near the grounding line, and 6-20 km
over most of the inland ice. The 2-D mesh is then vertically extruded using 10 equally
spaced, terrain following layers.

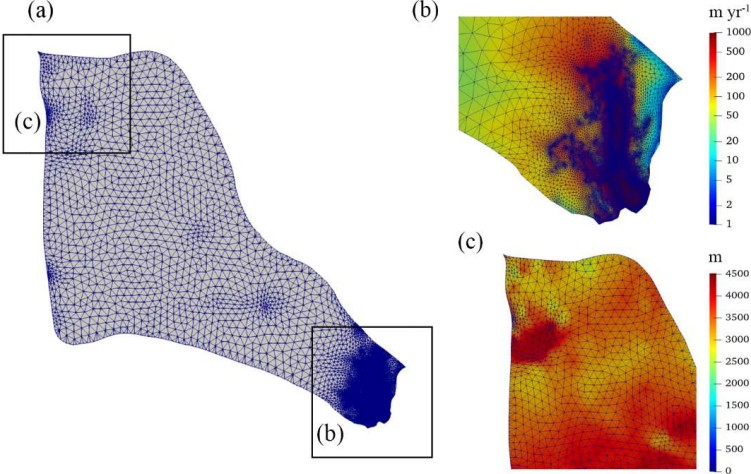

Fig. 3. The refined 2-D horizontal domain footprint mesh (a). Boxes outlined in (a) are shown in
detail overlain with surface ice velocity in (b) and with ice thickness in (c).
**3.2 Boundary Conditions**
The ice surface is assumed to be stress-free. At the ice front, the normal stress under the
sea surface is equal to the hydrostatic water pressure. On the lateral boundary, the
normal stress is equal to the ice pressure applied by neighboring glaciers and the normal
velocity is assumed to be 0. The bed for grounded ice is assumed to be rigid,
impenetrable, and fixed over time. For simplicity, we ignore the existence of Lake
Vostok and replace the lake with bedrock. We do this to avoid having to implement a
spatially variable sea level in our model, as the level of hydrostatic equilibrium in Lake
Vostok is several thousand meters higher than in the ocean. Our inverted drag
coefficient over the lake is very low, indicating that our simplification has only a small
influence on ice flow. However, our basal melt rates over the lake are probably
inaccurate, as we assume that geothermal flux from the lake bottom is applied directly



to the ice base, without accounting for circulation within the lake.
A linear sliding law is used to describe the relationship between the basal sliding
velocity and the basal shear force, on the bottom of grounded ice,

$$\tau_b = C \cdot u_b, \tag{1}$$

To avoid non-physical negative values, $C = 10^\beta$ is used in the simulation. We call $\beta$
the basal friction coefficient. $C$ is initialized to a constant value of $10^{-4}$ MPa m$^{-1}$ yr
(Gillet-Chaulet et al., 2012), and then replaced with the inverted $C$ in subsequent
inversion steps.
We relax the free surface of the domain by a short transient run to reduce the non-
physical spikes in initial surface geometry (Zhao et al., 2018). The transient simulation
period here is 0.5 yr with a timestep of 0.01 yr.
Following the same method as Kang et al. (2022), we improve the parameterization of
$\beta$ via $C$ in Eq 5 (Section 3.2.2) by considering basal temperature $T_{bed}$,

$$\beta_{new} = \beta_{old} + \alpha(T_m - T_{bed}), \tag{2}$$

where $\beta_{old}$ is from the inverse model, $\alpha$ is a positive factor to be tuned, $T_m$ is pressure
melting temperature. We take $\alpha$ to be 1, and use the parameterization of $\beta_{new}$ in Eq 1
in all the simulations (Kang et al., 2022). Using Eq 2 does not change simulated surface
velocities in the interior region.

**3.3 Basal Melt Rate**
Based on the inverted basal velocity and basal shear stress, we can calculate the basal
friction heat. We then produce the basal melt rate using the thermal equilibrium as
follows (Greve and Blatter, 2009):

$$M = \frac{G + \vec{u}_b\vec{\tau}_b + k(T)\frac{dT}{dz}}{\rho_i L}, \tag{3}$$

where $M$ is the basal melt rate, $G$ is GHF, $\vec{u}_b\vec{\tau}_b$ is the basal friction heat, $-k(T)\frac{dT}{dz}$ is the
upward heat conduction, $\rho_i$ is the ice density, and $L$ is latent heat of ice melt. GHF and
frictional heating from basal slip warm the base, while the upward heat conduction to
the interior cools the base.
**4 Simulation Results**
**4.1 Ice Velocity**
The modeled surface velocity fields with different GHFs are all very close to the
observed as expected by design of the minimization of misfit between the modeled and
the observed surface velocity in the inverse model. Therefore, we show only the Martos
et al. (2017) result as a representative example of all simulated velocity fields (Fig. 4).





The surface speed can reach as high as about 1000 m yr$^{-1}$ on the ice shelf (Fig. 4a, b).

Fig. 4c shows the modeled basal ice velocity. The modeled basal ice velocity is close to
0 in most of the inland region. The fast basal velocity in the middle of the region (Fig.
4c) is associated with subglacial canyon features (Fig. 1c), high basal temperature (Fig.
5) and small friction coefficient. In the grounded fast flow region, the basal ice velocity
can reach a maximum of 500 m yr$^{-1}$.

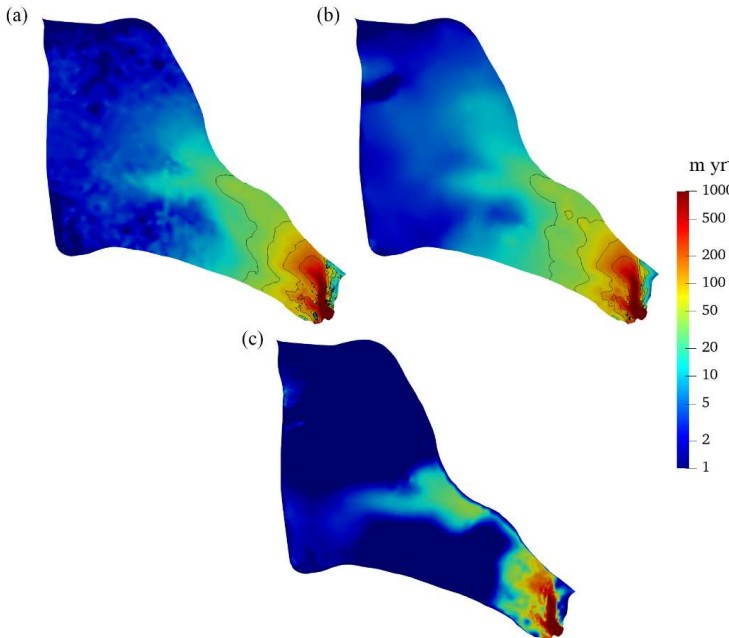


Fig. 4. (a) Observed surface velocity, (b) modeled surface velocity, (c) modeled basal velocity in the
experiment using the Martos et al. (2017) GHF. The black solid lines in (a) and (b) represent speed
contours of 30, 50, 100 and 200 m yr$^{-1}$.

**4.2 Basal Ice Temperature, Basal Friction Heat and Heat Conduction**
Fig. 5 shows the modelled basal temperatures from the five experiments. In the fast-
flowing region (defined as having surface speeds higher than 30 m yr$^{-1}$), the modelled
ice basal temperatures are all at the pressure melting point ("warm"). However, in the
slow-flowing region, the modeled ice basal temperature shows large difference between
GHF fields. In the experiment using the Martos et al. (2017) GHF (Fig. 5a), which has
the highest GHF over the domain, we get the largest area of warm base extending to all
but the inland southeast corner. The experiment using Shen et al. (2020) GHF (Fig. 5b),
which has the second highest GHF, yields the second largest area of warm base. The
experiment using Purucker et al. (2013) GHF (Fig. 5e), with the lowest GHF has the
smallest warm base area, which is mostly confined to the fast-flowing region.  All
experiments show cold basal temperatures in the southwest corner which is associated
with relatively thin ice above subglacial mountains (Fig. 1c).

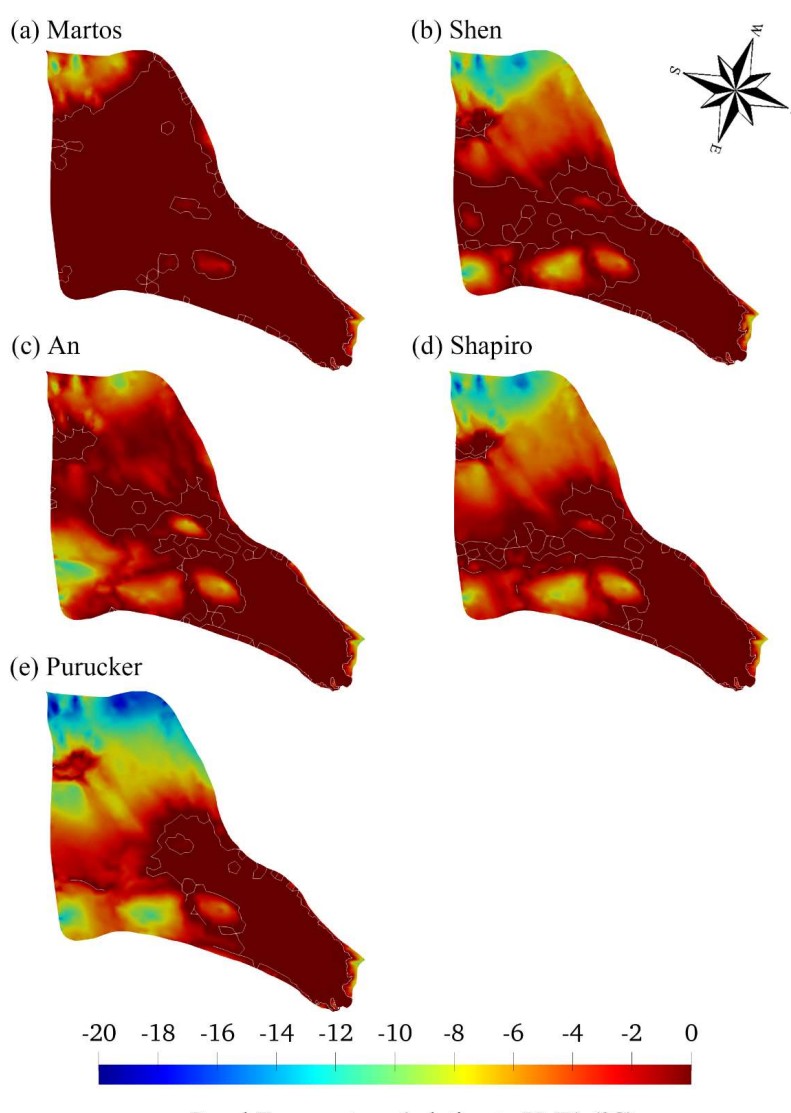


Fig. 5. Modelled basal temperature relative to pressure melting point, (a) to (e) corresponding to the
GHF (a) to (e) in Fig. 2. The ice bottom at the pressure-melting point is delineated by a white contour.


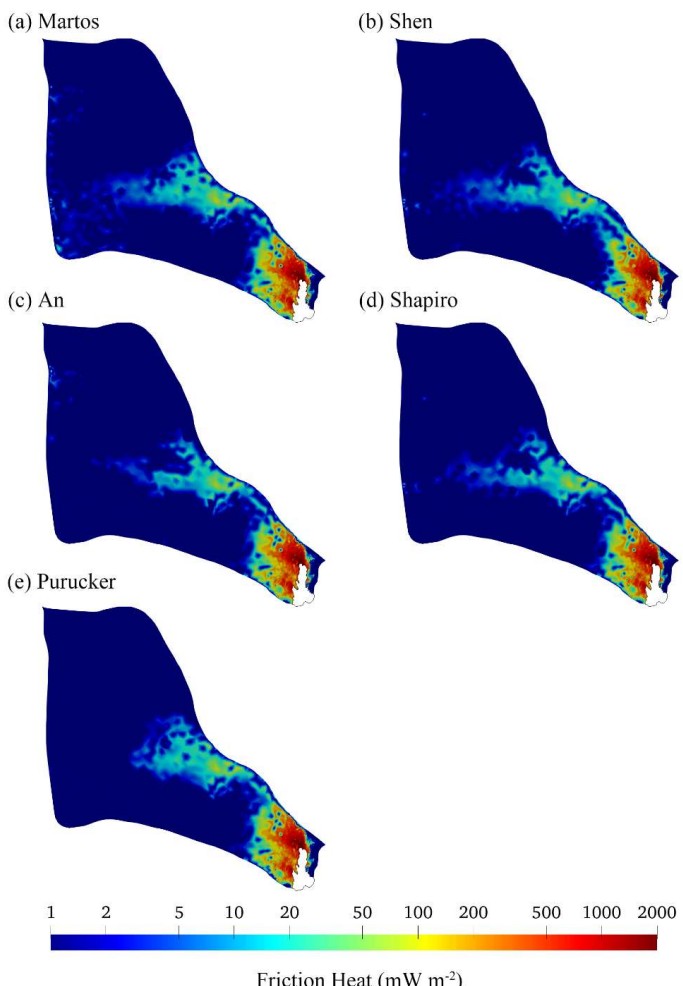

Fig. 6. Modelled basal friction heat.

The distribution of modeled basal friction heat is closely associated with that of modelled basal velocity. The patterns of basal friction heat with different GHFs are very similar in fast flow region, but have some differences in the middle of the domain (Fig. 6) where modelled basal velocity ranges between 5-20 m yr$^{-1}$ (Fig. 4).

The modelled basal friction heat is close to 0 where the surface ice velocity is less than 10 m yr$^{-1}$, but ranges widely by 10-2000 mW m$^{-2}$ elsewhere. Basal friction heating larger than 100 mW m$^{-2}$ occurs where surface velocity is more than 50 m yr$^{-1}$ and basal velocity is higher than 10 m yr$^{-1}$ (Fig. 6; Fig. 4), and it is then the dominant heat source.

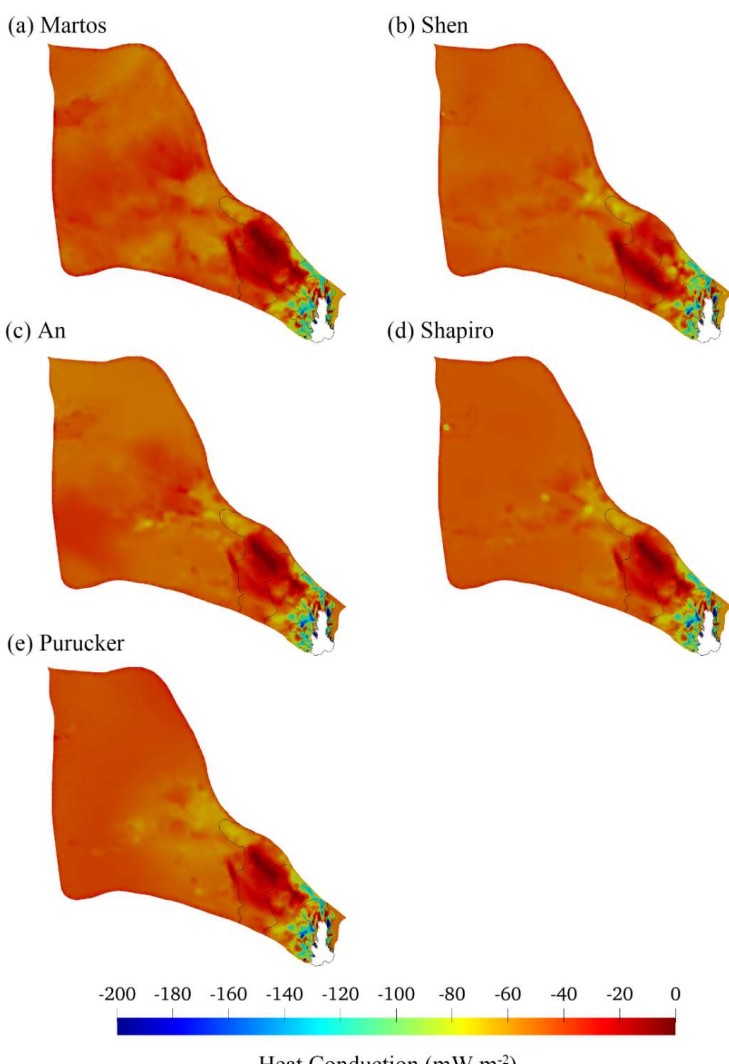

248

Fig. 7. Modelled heat change of basal ice by upward englacial heat conduction. The negative sign
means that the upward englacial heat conduction causes heat loss from the basal ice as defined by
the color bar with cooler colors representing more intense heat loss by conduction. (a) to (e)
corresponding to the GHF (a) to (e) in Fig. 2. The black solid curves represent modelled surface
speed contours of 30, 50, 100 and 200 m yr$^{-1}$, as in Fig. 4.

Fig. 7 shows the modeled heat change of basal ice by upward englacial heat conduction
in the five experiments. In the slow-flowing region where basal temperature is below
the pressure melting point, the upward basal heat conduction equals the GHF (Fig. 5,
Fig. 7). In the region where basal temperature reaches pressure melting point (Fig. 5)



with low basal velocity (Fig. 4c) and thick ice (≥2500 m; Fig. 1c), the heat loss caused
by upward basal heat conduction is < 30 mW m$^{-2}$ in all experiments (Fig. 7), reflecting
the development of a temperate basal layer that limits the basal thermal gradient. In the
fast-flowing tributaries with high basal velocity (Fig. 4c) and ice thickness <2000 m,
the heat loss caused by upward basal heat conduction can be very large, 100-200 mW
m$^{-2}$ near the grounding line (Fig. 7).

**4.4 Basal Melt Rate**
We calculate basal melt rate using the thermal balance equation (Eq 3). There are
significant differences in the five experiments due to large variability in GHF (Fig. 8).
The Martos et al. (2017) and then Shen et al. (2020) yield the largest areas with basal
melting. The experiments using An et al. (2015), Shapiro and Ritzwoller (2004) and
Purucker et al. (2013) yield similar total basal melting areas but have different spatial
patterns.

In most of the warm based regions, the modeled basal melting rate is <5 mm yr$^{-1}$ (Fig.
8) and basal friction heat is < 50 mW m$^{-2}$ (Fig. 6). Basal melting rates > 5 mm yr$^{-1}$ occur
with surface velocities > 100 m yr$^{-1}$ (Fig. 4, Fig. 8), where the basal friction heat is the
dominant heat source. In particular, the modeled basal melting rate is 50-400 mm yr$^{-1}$
in the two fast flow tributaries feeding the ice shelf that have surface velocities > 200
m yr$^{-1}$, and where the basal friction heat can reach 500-2000 mW m$^{-2}$ (Fig. 4, Fig. 6, Fig.
8). This is consistent with the findings of Larour et al. (2012) and Kang et al. (2022),
that the slow-flowing ice is more sensitive to GHF while the fast-flowing region is more
sensitive to basal friction heat.

There is relatively high modelled basal melt rate (4-10 mm yr$^{-1}$) localized at the central
subglacial canyon (Fig. 8, Fig. 1c), which is captured by all five GHF experiments, and
also consistent with the high values (0.5-1.0) of specularity content data there (Fig. 9).
Dow et al. (2020) found that the specularity content is a useful proxy for both water
depth and water pressure in regions of distributed water in subglacial canyons.

There is a location with modelled refreezing (negative melting rate) at the central
subglacial canyon, near the observed subglacial lake, in all five GHF experiments (Fig.
8). The value of specularity content there is low as 0-0.1 (Fig. 9), and freeze on is driven
by the steep topography around the canyon.

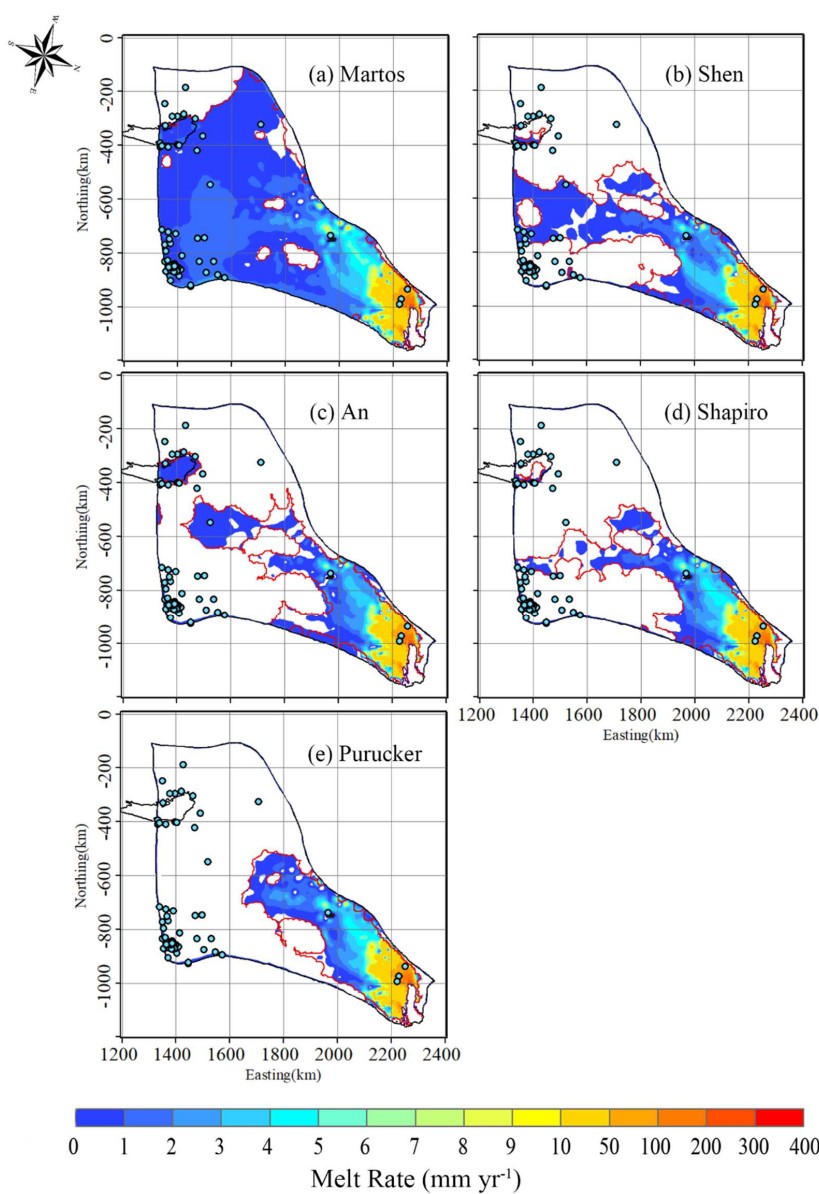

Fig. 8. Modelled basal melt rate, (a) to (e) correspond to the GHF (a) to (e) in Fig. 2. The ice bottom at pressure-melting point is surrounded by a red contour. The black curve denotes Lake Vostok. Stable subglacial lakes are shown as blue-green points with black circles. There is modelled basal refreezing at the central canyon painted in black.

**4.5 Evaluation of modelled results with 5 GHFs**

We use the locations of the observed subglacial lakes and specularity content to





discriminate between modeled basal melting (Fig. 8). Ideally, we would like to have a modeled ice base that is cold and dry where subglacial lakes do not exist and the specularity content is low, and a modeled ice base that is at the melting point where lakes and high specularity content are observed. In other words, we would like to use the available data to form a two-sided constraint that can penalize the model for being both too warm and too cold. If we only have a one-sided constraint, then we would always end up concluding that either the warmest or the coldest GHF map is best, regardless of whether that map was a reasonable representation of the basal state.

Observations of subglacial lakes are mostly a one-sided constraint on the basal thermal state. This is because lakes are only detectable if subglacial water accumulates in depressions that are deep compared to the radar wavelength and wide in comparison to the horizontal resolution of the radar system. Other forms of distributed hydrology, such as linked cavities or saturated subglacial sediments, do not produce the classic flat bright reflectors characteristic of subglacial lakes. Thus, the lack of observed subglacial lakes in a particular region cannot be taken as evidence that there is no subglacial water there. The mesh resolution of our model inland is about 20 km (Fig. 3). But 84% of the subglacial lakes have along-radar track lengths below 5 km, 94% are below 10 km, with only 5 lakes including Lake Vostok above 10 km (Fig. 9f). So the subglacial lakes may be too small for the ice model to resolve. Nonetheless, we compare our modeled basal thermal state with the observed locations of subglacial lakes. These comparisons show that all the experiments can capture all four subglacial lakes in the fast-flowing region (Fig. 8). But their performance in covering subglacial lakes in the slow-flowing region differ greatly.

In addition to the subglacial lakes, we use specularity content to derive a two-sided constraint on basal thermal state. Specularity content is an inherently noisy measure, so it is smoothed to 1 km along track values, and furthermore it is not unambiguously an indicator of wet beds. For example, specularity content is low in the fast-flowing region (Fig. 9, Fig. 4), where there must be lubricating water at the bed. Similar specularity results were also seen by Schroeder et al. (2013) for Thwaites Glacier, where high specularity values are seen under the major tributaries and the upstream trunk, but significant lower values of specularity in the fast-flowing region. This counter-intuitive result may be due to distinct morphologies and radar scattering signatures between water distributed in widespread subglacial conduits and water concentrated in just a few subglacial channels. Because of this effect, we only use the specularity content outside the fast-flowing region (defined as surface speed>30 m a$^{-1}$, Fig. 9).

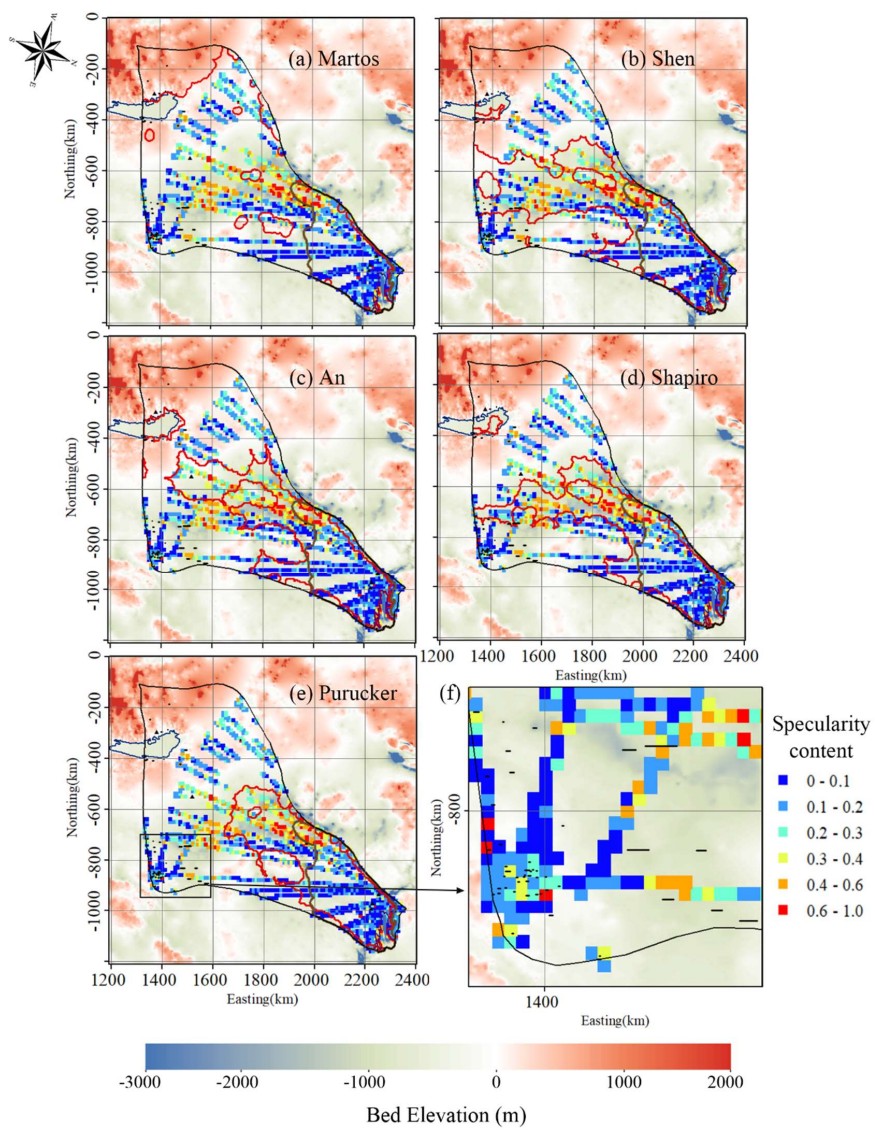

Fig. 9. Locations of specularity content (colored points) derived from radar data collected by ICECAP (Dow et al., 2020) and interpolated to 10 km by 10 km grids under the background of bedrock elevation. Specularity content > 0.4 indicates the likely presence of basal water. The ice bottom at pressure-melting point is surrounded by a red contour, (a) to (e) correspond to the five GHF maps (a) to (e) in Fig. 2. Lake Vostok is outlined by a blue curve. The brown curve is the contour of surface speed of 30 m a$^{-1}$. Subglacial lakes are shown at observed positions as a line segment of their length. Plot (f) is a zoom of the box in plot (e).


The specularity content data calculated from ICECAP survey lines suggests hundreds
of locations with basal water (Dow et al., 2020). The default resolution of specularity




content along the flight lines is 1 km (Dow et al., 2020), which is smaller than our model
resolution of 6-20 km in the slow flowing region. Water may accumulate in just a small
fraction of the grid cell even if the majority of the cell is warm because of water flow.
For comparability, with our simulation resolution we aggregated the specularity content
data onto 10 km by 10 km windows (Fig. 9). The 10 km window is a somewhat arbitrary
choice, but smaller windows (we tried 2 and 5 km) reduce the data available and noise
becomes larger, while larger windows (we tried 15 and 20 km) restrict spatial resolution.
We then take the upper fifth percentile of the specularity content, $specularity_5$ of each
window as a water indicator rather than its mean value to allow for localized water
collection or unfavorable bed reflection geometry, while also excluding spurious signals
in the noisy specularity data. Young et al. (2016) suggested that specularity larger than
0.4 was an indicator of a warm bed. This is also consistent with the largest subglacial
lake in the domain with length of 28 km having specularity content>0.4 (Fig. 9f). There
are also some smaller lakes (several km along-track lengths) with specularity content
between 0.2 and 0.4, so a warm threshold of 0.4 would not capture these features.  The
cold threshold need not be the same as the warm bed one, and so we explored different
values for cold thresholds of 0.2, 0.3, 0.4, but found that the 0.2 cold threshold provided
best discrimination between models, and also maximizes the available data.
To evaluate modelled basal conditions with specularity content, we define a warm hit
rate as the ratio of the number of grid cells with modelled warm bed that have
$specularity_5 > 0.4$ to the total number of grids with $specularity_5 > 0.4$.  Similarly, cold
hit rate is defined as the ratio of the number of grid cells with $specularity_5 < 0.2$.
One simple measure of quality is just the average of warm hit rate and cold hit rate, but
we also want an unbiased evaluation of GHF to have similar capabilities in capturing
both warm bed and cold bed regions. Therefore, we define *imbalance* as

$$imbalance = \frac{warm\ hit\ rate - cold\ hit\ rate}{warm\ hit\ rate + cold\ hit\ rate},$$

as it reflects the difference between warm hit rate and cold hit rate, and has a value
between -1 and 1. The closer to zero *imbalance* is, the more confidence we have in the
model result. The overall performance is estimated by averaged hit rate minus the
absolute value of *imbalance*.
The Martos GHF has the highest warm hit rate and the lowest cold hit rate since it has
the largest modelled warm bed area. The averaged hit rates of modelled results with 5
GHF are very close, with differences < 0.13 (Table 3). The Shapiro, Purucker, then Shen
have the highest averaged hit rate using all the values for threshold of cold bed, and the
differences between their averaged hit rate < 0.04.
Martos and Shen have positive *imbalance*, which means that their warm hit rate is
higher than their cold hit rate. In contrast, An, Shapiro and Purucker have negative
*imbalance*. Martos has the largest *imbalance* because its warm hit rate overwhelms its
cold hit rate. The absolute *imbalance* of Shen is < 0.05 with all three cold hit thresholds

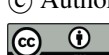

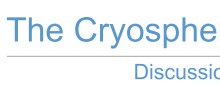

we used and always the smallest (Table 3) of the GHF. The Shapiro absolute *imbalance*
the second smallest with all the cold hit thresholds. Therefore, Shen and Shapiro rank
the top two according to *imbalance* between warm hit rate and cold hit rate.
Considering the overall performance by averaged hit rate minus the absolute value of
*imbalance*, Shen is the best, Shapiro the second, Purucker the third, An the fourth and
Martos the last (Table 3). The ranking is robust with all three cold hit thresholds.
Table 3. Warm hit rate, cold hit rate, averaged hit rate, imbalance and overall
performance for the modelled results with 5 GHFs. The threshold of *specularity$_5$* is
taken as 0.4 for warm hit rate, and 0.2 for cold hit rate.

| GHF | warm hit rate | cold hit rate | averaged hit rate | Imbalance | averaged hit rate – abs(imbalance) |
|---|---|---|---|---|---|
| Martos | 0.9560 | 0.1648 | 0.56 | 0.71 | -0.15 |
| Shen | 0.6588 | 0.6564 | 0.65 | 0.0018 | 0.65 |
| An | 0.4340 | 0.7652 | 0.60 | -0.28 | 0.32 |
| Shapiro | 0.5975 | 0.7822 | 0.69 | -0.13 | 0.56 |
| Purucker | 0.5283 | 0.8201 | 0.67 | -0.22 | 0.45 |


## 5 Discussion

Wright et al. (2012) modelled basal temperature of Totten Glacier using the Glimmer
ice sheet model with a constant GHF of 54 mW m$^{-2}$. Their modelled area of basal warm
ice is between what we simulated using Martos et al. (2017) and Shen et al. (2020),
covering most of the lakes and lake-like features but missing some near Lake Vostok.
Dow et al. (2020) ran the Ice Sheet System Model (Larour et al., 2012) with a constant
GHF of 55 mW m$^{-2}$, producing a warm bed region slightly larger than we simulated
using the Shen et al. (2020) GHF (which has a mean of 58 mW m$^{-2}$ in this region, Table
2). Eisen et al. (2020) modeled the basal temperature of Antarctic ice sheet with the
Parallel Ice Sheet Model using four different GHF datasets (Shapiro and Ritzwoller,
2004; Fox Maule et al., 2005; An et al., 2015; Martos et al., 2017). The mean modelled
basal temperature of the different GHFs appear close to our result using the Shen et al.
(2020) GHF, with basal temperatures reaching the pressure melting point in the fast
flow region and the central upstream region of Totten Glacier.
Kang et el. (2020) evaluated basal thermal conditions underneath the Lambert-Amery
glacier system using six GHFs, and found that the two most recent GHF fields inverted
from aerial geomagnetic observations and which have the highest GHF values,
produced the largest warm-based area, and best matched the observed distribution of
subglacial lakes. This might be expected as there was only a one-sided constraint used,
and warm based models produced matches with more lakes.
Although the basal ice in fast-flowing regions is all at pressure melting point because
basal friction heat dominates the heat balance, the modelled basal melt rate of the





grounded ice in fast-flowing regions exhibits large differences across-models. The
modelled basal melt rate is associated with the modelled basal friction heat, which is a
function of the modelled basal velocity and basal shear stress, the accuracy of which
depends on the configuration and constraints of the ice sheet model used. Our modelled
maximum basal melt rate on the grounded ice is 0.4 m yr$^{-1}$ near the grounding line. This
is close to the modelled maximum basal melt rate of 0.34 m yr$^{-1}$ near the grounding line
by Dow et al. (2020), where they calculated the basal melt rates as a function of
combined GHF and frictional heating using the Ice Sheet System Model. We know of
no observations of the basal melt rates of grounded ice in Totten Glacier.
Modelled basal sliding speeds by Dow et al. (2020) range from 0.06 m yr$^{-1}$ inland to
900 m yr$^{-1}$ at the grounding line, which is close to our result (Fig. 4). Dow et al (2020)
simulate basal sliding generally where bedrock is below sea level, with an area close to
our simulation with a basal sliding coefficient $\beta_{old}$ and which is larger than ours using
the improved basal sliding coefficient $\beta_{new}$ (Eq 2) found by considering the basal
temperature relative to pressure-melting point. The modelled basal sliding speed
reaches a local maximum at the middle of the subglacial canyon system (Fig. 4), which
leads to local maxima in basal friction and basal melt rate (Fig. 8), and is consistent
with the high values of specularity (Fig. 9).
To evaluate the simulation results, we compare the simulated basal melting area with
the locations of the discovered subglacial lakes and specularity content derived from
radar data collected by ICECAP (Dow et al., 2020). Specularity is a parameterization
that estimates the along-track angularly narrow component of bed echo energy
compared with the isotropic diffuse energy component (Schroeder et al., 2015).
Specularity is determined by a set of ice/bed properties including the length, width and
thickness of the water body, its conductivity, and the roughness of the ice/water
interface. Off-nadir across-track reflectors may also produce glints creating noise in the
specularity distribution. Hence, interpretation of specularity is ambiguous and
dependent on the local bed morphology. This led us to experiment with a range of
windows over which to aggregate the bed reflection energy, and various thresholds for
estimating cold and warm beds. We were able to use the numerous subglacial lakes in
the region as a guide to setting these parameters, bearing in mind that the observations
of subglacial lakes are a one-sided constraint. If the modeled basal melting area misses
the subglacial lake or high specularity content, the model is underestimating the basal
temperature at that location. However, if the basal melting is simulated in areas without
observed subglacial lakes, it is unclear if this is because the models overestimate the
temperature in those areas, or if the water under the ice sheet has not been detected.
Moreover, a hypersaline lake and various other water saturated environments seem to
exist below cold ice beneath Devon Island ice cap in Canada (Rutishauser et al., 2022).
In addition, relatively high electrical conductivity beds like water saturated clays can
lead to false positives in radar detections of subglacial water bodies (Talalay et al.,
471  2020).





Our evaluation using specularity content is a two-sided constraint and thus improves on
observed subglacial lakes as a discriminating feature of cold and warm beds. The
experiment with Martos et al. (2017) GHF models the largest region of basal melt, and
covers most observed subglacial lake locations. However, it ranks worst in the
evaluation using specularity content, because it cannot capture cold beds well.
**6 Conclusions**
In this study we diagnose the basal thermal state of Totten Glacier by coupling a forward
model and an inverse model and using five different GHFs. By comparing modelled
basal temperature distributions with metrics derived from specularity content data we
evaluate the reliability of the five GHF data in this area.
We find there are significant differences in the spatial distributions of modelled
temperate ice with different GHFs, and the differences are mainly concentrated in the
slow ice flow regions. The modelled basal thermal state (frozen/melting) in the slow
ice flow region is mainly determined by the heat balance between GHF and englacial
upward heat conduction, and the basal melting rate is generally less than 5 mm yr$^{-1}$.
However, there is local maximum in modelled basal melt rate (4-10 mm yr$^{-1}$) at the
central subglacial canyon, which could be explained by the local high basal sliding
velocity and frictional heat that are captured by all GHF experiments. This is consistent
with the high values of specularity content data there.
The basal heat balance in the fast ice flow region is mainly determined by the basal
frictional heat. The basal ice in the fast flow region is all at the melt point. The modeled
basal melting rate is 50-400 mm yr$^{-1}$ in the two fast flow tributaries feeding the ice shelf
with surface velocity greater than 200 m yr$^{-1}$, where the basal friction heat is 500-2000
mW m$^{-2}$.
Our evaluation using specularity content as a two-sided constraint, gives quite different
result than only using observed locations of subglacial lakes. Simulations with the
Martos et al. (2017) GHF yields the largest region of basal melt, which covers most
observed subglacial lake locations, however, its cold bed fit with specularity content is
poor and shows huge imbalance in modelling warm bed and cold bed regions. Overall,
Martos et al. (2017) GHF ranks last in the evaluation with specularity content. Shen et
al. (2020) GHF yields the second largest area of basal melt and second best agreement
with the locations of the subglacial lakes, and also scores well in modelling both warm
and cold bed areas.  Shen et al. (2020) GHF and Shapiro and Ritzwoller (2004) GHF
rank the top two according to the evaluation with specularity content. The best-fit
simulated result shows that most of the inland bed area is frozen. Only the upstream
subglacial canyon inland reaches pressure-melting point, and modelled basal melting
rate there is 0-10 mm yr$^{-1}$.
**Data availability**
MEaSUREs    BedMachine    Antarctica,    version    2,    is    available    at



https://doi.org/10.5067/E1QL9HFQ7A8M (Morlighem, 2020). MEaSUREs InSAR-
based Antarctic ice velocity Map, version 2, is available at
https://doi.org/10.5067/D7GK8F5J8M8R (Rignot et al., 2017). MEaSUREs Antarctic
Boundaries for IPY 2007–2009 from Satellite Radar, version 2 is available at
https://doi.org/10.5067/AXE4121732AD (Mouginot et al., 2017). The subglacial lake
dataset is available at https://doi.org/10.1038/s43017-021-00246-9 (Livingstone et al.,
2022). The specularity content dataset https://doi.org/10.5281/zenodo.3525474 (Dow
et al., 2020). ALBMAP v1 and the GHF dataset of Shapiro and Ritzwoller (2004) are
available at https://doi.org/10.1594/PANGAEA.734145 (LeBrocq et al., 2010b). The
GHF dataset of An et al. (2015) is available at
http://www.seismolab.org/model/antarctica/lithosphere/AN1-HF.tar.gz (last access: 11
April 2023). The GHF dataset of Shen et al. (2020) is available at
https://sites.google.com/view/weisen/research-products?authuser=0 (last access: 11
April 2023). The GHF dataset of Martos (2017) is available at
https://doi.org/10.1594/PANGAEA.882503. The GHF dataset of Purucker (2012) is
available at http://websrv.cs.umt.edu/isis/index.php/Antarctica_Basal_Heat_Flux (last
access: 11 April 2023). The modelled basal temperature, basal melt rate and the upper
fifth percentile of the specularity content in this paper will be available at
https://doi.org/10.5281/zenodo.7825456.

**Author contributions.**
LZ and JCM conceived the study. LZ, MW, and JCM designed the methodology. YH,
LZ, and YM carried out the simulations and produced the estimates and figures. LZ
wrote the original draft, and all the authors revised the paper.

**Competing interests.**
The authors declare no conflict of interest.

**Acknowledgments**
This work was supported by the National Natural Science Foundation of China (No.
41941006), National Key Research and Development Program of China
(2021YFB3900105), State Key Laboratory of Earth Surface Processes and Resource
Ecology (2022-ZD-05) and Finnish Academy COLD Consortium (No. 322430).

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
