# Peer review of "Using specularity content to evaluate five geothermal heat flux maps of 2 Totten Glacier Yan Huang1, Liyun Zhao1,2\*, Yiliang Ma1, Michael Wolovick3, John C. Moore1,4\* 3 1 College of Global Change and Earth System Science, Beij"

_The Cryosphere, 2023_

## Referee Comment (RC2)

[referee-annotated manuscript omitted]

---

## Author Comment (AC2)

Referee's comments are in red, our reply on black, quotes in the revised manuscript in blue.

The paper takes a thoughtful approach to assess a collection of Geothermal Heat Flux (GHF) using two sets of radar sounding observations: the detection of subglacial lakes and bed echo specularity content. The authors rightly point out that "one-sided" tests using either observable will result in the selection of the highest GHF values. The authors are thoughtful about where they do and do not apply the specularity content in terms of upstream and downstream portions of the catchment. The authors are also thoughtful about the difference between how the lakes and specularity observations are used in terms of "one-sided" vs. "two-sided" constraint.

However, the authors seem to view the choice as either a "one-sided" approach that only evaluates if the lakes/high-specularity correspond to thawed areas or a "two-sided" approach in which that comparison is combined with evaluation of "no lake"/low-specularity correspond to cold areas. However, it seems like there's another option. To "reward" the match between lakes/high-specularity and thawed areas as in the "one-sided" and then to "penalize" a mismatch between lakes/high-specularity and cold areas. This seems like more than the "one-sided" approach and like it might not be vulnerable to the same preference for the highest meld as the pure "one-sided" approach. However, it also seems like it has the benefit of applying to both lakes and specularity. It has the additional benefit of allowing the specularity to be used in all regions of the catchment. This seems additionally important because, just as the authors describe for lakes, it's not the case that low-specularity areas have to correspond to cold areas, it can correspond to thawed areas where water is not pooled in sufficient quantities to be specular and/or form a lake. As a result, this intermediate between "one-sided" and "two-sided" metrics could apply to both observables.

Reply: We tried the option that the referee suggested, i.e., to "reward" the match between lakes/high-specularity and thawed areas as in the "one-sided" and then to "penalize" a mismatch between lakes/high-specularity and cold areas.

The reward of the match between lakes/high-specularity and thawed areas is the ***warm hit rate*** we defined as the ratio of the number of grid cells with modelled warm bed that have $specularity_5 > 0.4$ to the total number of grids with $specularity_5 > 0.4$.

We name the penalize of mismatch between lakes/high-specularity and cold areas as ***cold miss-fit rate***, which can be defined as the ratio of the number of grid cells with modelled cold bed that have $specularity_5 > 0.4$ to the total number of grids with $specularity_5 > 0.4$.

We call the new metrics that the referee suggested ***Total rate 2***, which is

$$\textbf{\textit{Total rate 2}} = \textit{warm hit rate} - \textit{cold miss-hit rate.}$$

For comparison, we call the "two-sided" metrics in the manuscript as ***Total rate 1***,

which is the *average* of warm hit rate and cold hit rate minus the abs(*imbalance*), where *imbalance* reflects the difference between warm hit rate and cold hit rate.

We show **Total rate 1** and **Total rate 2** in the table below using the 5 GHF datasets.

Table 1. *Warm hit rate*, *cold hit rate*, *cold miss-fit rate*, *Total rate 1* and *Total rate 2* for the modelled results with 5 GHFs. The threshold of *specularity$_5$* is taken as 0.4 for warm hit rate, and 0.2 for cold hit rate.

| GHF | Warm hit rate | Cold hit rate | Cold miss-hit rate | Total rate1 (original result) | Total rate 2 |
|---|---|---|---|---|---|
| Martos et al., 2017 | 0.9750 | 0.1042 | 0.019 | -0.27 | 0.956 |
| Shen et al., 2020 | 0.7250 | 0.5682 | 0.256 | 0.53 | 0.469 |
| An et al., 2015 | 0.5563 | 0.6591 | 0.425 | 0.52 | 0.1313 |
| Shapiro and Ritzwoller, 2004 | 0.5750 | 0.6951 | 0.406 | 0.54 | 0.169 |
| Purucker, 2012 | 0.5375 | 0.7254 | 0.444 | 0.48 | 0.0935 |

We can see the ranking of **Total rate 2** is the same as that using *Warm hit rate*. Especially for the highest GHF, Martos et al. (2017), the area of modelled warm bed is much larger than modelled cold bed, hence the value of *warm hit rate* is much larger than *cold miss-fit rate,* and plays the dominant role in **Total rate 2.**

Therefore, the *cold miss-fit rate* did little to "penalize" the mismatch between lakes/high specificity and cold areas. It cannot be serve as an intermediate between "one-sided" and "two-sided" metrics. We prefer to stick on the "two-sided" metrics used in the manuscript.

---

## Author Comment (AC3)

Referee's comments are in red, our reply in black, quotes in the revised manuscript in blue.

Dear Authors, dear Editor,

At this stage, the manuscript needs some major edits.

The main goal of the manuscript is to evaluate GHF data sets in the Totten region using specularity results. The idea and the novelty of the method in this region are laudable and of interest, as from one data set to another, GHF values vary a lot. However, a more detailed review of the literature, statistical work, .... are needed to ascertain what the authors suggest at the moment. In addition, the whole manuscript requires technical revision work (disjointed paragraphs, figures, figures captions, ... ). There are key GHF data sets that are missing from your analysis and the seminal reference of Burton-Johnson on *Review article: Geothermal heat flow in Antarctica: current and future directions* https://tc.copernicus.org/articles/14/3843/2020/ is not referenced.
Reply: Thank you for your encouraging general comment. We add 3 new GHF datasets that were missing and also cite the above review article as a reference. We analyze the new results, do the statistical work, and improve the tables and figures as the referee suggested.

I would therefore suggest the following actions (general comments):

1) Describe and use all available datasets published, especially since your aim is to do a ranking of the various datasets. Doing so would add great value to your paper, especially since the missing two datasets are quite recent. However, the idea of doing a ranking of the GHF data set is also tricky in my opinion, as it depends strongly on the model resolution, the ranking parameters chosen, .... For example in your paper, you mention that the Shen et al, 2020 data set is "the best", while we can clearly see that the Martos et al, 2017 data set matches well for "warm conditions". Also, the grid resolution of all these data sets are critical in your analyses and in consequence they have to be at least listed in table 2.

The key references for the missing GHF data sets are : 1) Haeger, Carina, A. G. Petrunin, and M. K. Kaban. "Geothermal heat flow and thermal structure of the Antarctic lithosphere." Geochemistry, Geophysics, Geosystems 23.10 (2022): e2022GC010501. 2) Lösing, Mareen, and J. Ebbing. "Predicting geothermal heat flow in Antarctica with a machine learning approach." Journal of Geophysical Research: Solid Earth 126.6 (2021): e2020JB021499. 3) Stål, Tobias, et al. "Antarctic geothermal heat flow model: Aq1." Geochemistry, Geophysics, Geosystems 22.2 (2021): e2020GC009428.

Reply: We agree that doing a ranking is tricky. We avoid saying "the best" in the revision. We add the 3 new GHF datasets that the referee suggested, and add the grid

resolution of all these datasets in Table 2. We also rearranged the order of listing the 8 GHF by their approach in the tables and figures.

All the GHF datasets are bilinearly interpolated into 2.0 km resolution. Then we calculated the ensemble mean of the eight GHF maps, and a uniform GHF value, 59 mW m$^{-2}$, which is the area average of ensemble mean (Table 2).

Table 2 The ten GHF map used with the mean, range and resolution in our region.

| GHF map | Method | Mean (mW m$^{-2}$) | Range (mW m$^{-2}$) | Resolution (km) |
|---|---|---|---|---|
| Martos et al., 2017 | airborne geomagnetic data derived model | 65 | 51-70 | 15 |
| Purucker, 2012 | satellite geomagnetic data derived model | 51 | 37-67 | 100-400 |
| Shen et al., 2020 | seismic model | 58 | 42-63 | 100-200 |
| An et al., 2015 | seismic model | 51 | 34-56 | 100-200 |
| Shapiro and Ritzwoller, 2004 | seismic model | 58 | 44-63 | ~100 |
| Stål et al., 2021 | multivariate approach | 60 | 34-80 | 20 |
| Lösing et al., 2021 | machine learning | 63 | 47-71 | 55 |
| Haeger et al., 2022 | multivariate approach | 64 | 54-67 | 10 |
| Mean GHF | Ensemble mean of the 8 datasets above interpolated into 2.0 km resolution | 59 | 48-61 | 2 |
| Constant GHF | mean of the ensemble mean GHF | 59 | 59 | 2 |

2) I strongly suggest a) adding statistical analyses to all parameters described (especially parameters shown on figure 5 to figure 7), perhaps stdev or another appropriate statistical parameter. b) adding a description of the bed roughness / topography influence. The bed topography influences the heat dissipation (difference between convex and concave bedrock shape).

Reply: We add ensemble mean and standard deviation (stdev) for the 8 GHF in updated Fig. 2. Then we do experiments using 10 GHF including the 8 GHF datasets, the ensemble mean GHF, and a constant GHF, which is the mean of the ensemble mean GHF. We calculate the stdev for the results (parameters) using 8 GHF datasets, and add new panels in the figures (see below).

We re-arranged the panels in the figures. We put the GHF derived by geomagnetic data in the first row, the GHF derived by seismic model the second row, the GHF derived by multivariate approach the third row, and the ensemble mean GHF, constant GHF, and standard deviation the last row.

[Figure]

Fig. 1. The spatial distribution of GHF listed in Table 2 over our domain (a)-(j). The ensemble mean GHF and standard deviation (SD) of the 8 GHF (a)-(h) are given in (i) and (k). Panel (j) shows the constant GHF of 59 mW m⁻². The purple line depicts the grounding line. The blue curve depicts Lake Vostok. The blue star denotes Dome C.

[Figure]

Fig. 5. Modelled basal temperature relative to pressure melting point, (a) to (j) corresponding to the GHF (a) to (j) in Fig. 2. Panel (k) is the standard deviation of 8 modelled basal temperatures (a)-(h). The ice bottom at the pressure-melting point is delineated by a gray contour. The purple line depicts the grounding line. The blue curve depicts Lake Vostok. The blue star denotes Dome C.

[Figure]

Fig. 6. Modelled basal friction heat, (a) to (j) corresponding to the GHF (a) to (j) in Fig. 2. Panel (k) is the standard deviation of 8 modelled basal friction heat (a)-(h). The purple line depicts the grounding line. The blue curve depicts Lake Vostok. The blue star denotes Dome C.

[Figure]

Fig. 7. Modelled heat change of basal ice by upward englacial heat conduction. The negative sign means that the upward englacial heat conduction causes heat loss from the basal ice as defined by the color bar with cooler colors representing more intense heat loss by conduction. (a) to (j) corresponding to the GHF (a) to (j) in Fig. 2. Panel (k) is the standard deviation of 8 modelled basal friction heat (a)-(h). The brown solid curves represent modelled surface speed contours of 30, 50, 100 and 200 m yr$^{-1}$, as in Fig. 4. The purple line depicts the grounding line. The blue curve depicts Lake Vostok. The blue star denotes Dome C.

[Figure]

Fig. 8. Modelled basal melt rate, (a) to (j) correspond to the GHF (a) to (j) in Fig. 2. The ice bottom at pressure-melting point is surrounded by a red contour. The black curve depicts Lake Vostok. Stable subglacial lakes are shown as blue-green points with black circles. The purple line depicts the grounding line. There is modelled basal refreezing at the central canyon painted in black.

[Figure]

Fig. 9. Locations of specularity content (colored points) derived from radar data collected by ICECAP (Dow et al., 2020) and interpolated to 10 km by 10 km grids under the background of bedrock elevation. Specularity content > 0.4 indicates the likely presence of basal water. The ice bottom at pressure-melting point is surrounded by a red contour, (a) to (j) correspond to the GHF (a) to (j) in Fig. 2. Lake Vostok is outlined by a blue curve. The brown curve is the contour of surface speed of 30 m a⁻¹. Subglacial lakes are shown at observed positions as a line segment of their length. Plot (k) is a zoom of the box in plot (h).

We also add a description of the bed roughness / topography influence on the heat dissipation as "The bed topography affects heat diffusion pathways to the earth's crust, therefore has influence on GHF at kilometer scales. Typically, near-surface temperature gradient is decreased near topographic rises and increased near topographic depressions (Bullard, 1938; Colgan et al., 2021)".

References:

Bullard, E. C.: The disturbance of the temperature gradient in the earth's crust by inequalities of height, Geophysical Supplements, Mon. Not. R. Astron. Soc., 4, 360–362, 1938.

Colgan, W., MacGregor, J. A., Mankoff, K. D., Haagenson, R., Rajaram, H., Martos, Y. M., Morlighem, M., Fahnestock, M. A., and Kjeldsen, K. K.: Topographic correction of geothermal heat flux in Greenland and Antarctica. J. Geophys. Res.-Earth, 126, e2020JF005598, https://doi.org/10.1029/2020JF005598, 2021.

3)  An interesting exercise (and fairly straightforward) to add to your work presented here would be to look at model simulation output with as input a single uniform GHF value for your whole domain of interest. This would allow you to have a base for comparaison and statistical analysis.

Reply: Yes, we calculated the ensemble mean of the 8 GHF datasets. Then we get the area average of this ensemble mean, 59 mW m$^{-2}$, and use it as a single uniform GHF value. We did experiments using 10 GHF including the 8 GHF datasets, the ensemble mean GHF, and the constant GHF. We expand the result analysis and figures correspondingly.

4)  The figures need to be clearer (see specific comments): e.g. fig 2 to 7, it would be useful to have a few geographic locations (Dome C, grounding line, Vostok, a few lat-lon) so readers can pinpoint where geographically the differences between datasets lie; and use a thicker black / white line, we do not see it at all on the figure panels (velocity contours, ice bottom at the pressure-melting point).

Reply: We revised Figure 2 to 7, see the reply above, making the figure clearer, adding the coordinates the same as Fig. 1, adding the geographic locations (Dome C, grounding line, Vostok), and using a thicker line for contours. Some updated figures are shown above, and some other updated figures are shown below.

[Figure]

Fig. 3. The refined 2-D horizontal domain footprint mesh (a). Boxes outlined in (a) are shown in detail overlain with surface ice velocity (unit: m yr⁻¹) in (b) and with ice thickness in (c). The while line in (a) and (b) depicts the grounding line. The black curve in (a) and (c) depicts Lake Vostok. The blue star denotes Dome C.

[Figure]

Fig. 4. (a) Observed surface velocity, (b) modeled surface velocity using the Martos et al. (2017) GHF, (c) modeled basal velocity in the experiment using the Martos et al. (2017) GHF. The brown solid lines in (a) and (b) represent speed contours of 30, 50, 100 and 200 m yr⁻¹. The purple line depicts the grounding line. The blue curve depicts Lake Vostok. The blue star denotes Dome C.

5) The figure captions need to be rewritten. It's unclear as is (see specific comments).

Reply: We rewrote the figure caption according to the referee's specific comments, see the reply above.

6) The bibliography as a whole has to be checked: some references are wrong, and some are not actually cited in the main document (see also specific comments). a) Please always mention the whole ref, i.e. Martos et al (2017) and not Martos in the text and also on the figures and tables. b) There are several references listed in the

bibliography that are never cited in the main text: e.g Stearns et al, 2008 ; Cuffey, K. M., and Paterson 2010 ; Van Liefferinge et al, 2018 ; Wolovick 2021b, ... These references have to be cited explicitly in the text, especially since they support the results of the manuscript. c) There are typos in some references: e.g Van Liefferringe and Pattyn, 2013. d) Wright and Siegert, 2012 is mentioned in the text but not in the linked table, ... it lacks consistency.

Reply: We double checked the bibliography. a) We mention the whole ref, i.e. Martos et al. (2017) and not Martos in the text and also on the figures and tables. b) We checked the references and cited the related references in proper places, and removed some unrelated references. c) We corrected the typos in some references. d) We add Wright and Siegert, 2012 in the updated Table 1.

....

I attach a detailed review of the paper for the specific line-by-line comments, see attached PDF.

Reply: Thank you for your detailed review. We revised the manuscript line by line according to your specific comments. We edited the words and sentences. For your comment to Line 167, "please mention what is the effect on the basal friction coefficient used", we addressed this in the original text in Line 169-171 as "Our inverted drag coefficient over the lake is very low, indicating that our simplification has only a small influence on ice flow". The most likely effect is that we have slightly too much drag over the lake (i.e., we have low but non-zero drag) compensated by slightly less drag outside of the lake.

For your comment to Line 374-381, i.e. "It is a simple but interesting comparison to evaluate GHF/specularity hit rate", we calculated it. Recall that

averaged hit rate = 0.5×(warm hit rate + cold hit rate),

overall performance = averaged hit rate − abs(*imbalance*).

The updated Table 3 is as follows. Shen et al. (2020) ranks the first, Shapiro and Ritzwoller (2004) the second, Purucker (2012) the third, An et al. (2015) the fourth, Martos et al. (2017), Stål et al. (2021), Lösing et al. (2021) and Haeger et al. (2022) get negative scores, and rank lowest among the 8 GHF (Table 3). The constant GHF produces a lower score than any of the published maps, and the gridded mean GHF map ranks below the top four individual maps, suggesting that at least in this domain that some models have significantly better skill than others.

Table 3. Warm hit rate, cold hit rate, averaged hit rate, imbalance and overall performance for the modelled results with eight GHFs in Table 2, ensemble mean GHF, and constant GHF of 59 mW m$^{-2}$. The overall performance is calculated by averaged hit rate minus the absolute value of imbalance. The threshold of *specularity$_5$* is taken as 0.4 for warm hit rate, and 0.2 for cold hit rate.

| GHF | warm hit rate | cold hit rate | averaged hit rate | Imbalance | overall performance |
|---|---|---|---|---|---|
| Martos et al., 2017 | 0.9560 | 0.1648 | 0.56 | 0.71 | -0.15 |
| Purucker, 2012 | 0.5283 | 0.8201 | 0.67 | -0.22 | 0.45 |
| Shen et al., 2020 | 0.6588 | 0.6564 | 0.65 | 0.0018 | 0.65 |
| An et al., 2015 | 0.4340 | 0.7652 | 0.60 | -0.28 | 0.32 |
| Shapiro and Ritzwoller, 2004 | 0.5975 | 0.7822 | 0.69 | -0.13 | 0.56 |
| Stål et al., 2021 | 0.8750 | 0.2405 | 0.56 | 0.57 | -0.01 |
| Lösing et al., 2021 | 0.9313 | 0.2216 | 0.58 | 0.62 | -0.04 |
| Haeger et al., 2022 | 0.9688 | 0.1458 | 0.56 | 0.74 | -0.18 |
| Mean GHF | 0.8750 | 0.4205 | 0.65 | 0.35 | 0.30 |
| Constant GHF | 0.9813 | 0.1042 | 0.54 | 0.81 | -0.27 |

The score of overall performance can be negative (Table 3). So we use the averaged hit rate which is positive, and calculated GHF/averaged hit rate for 8 GHF as shown in the table below. We can see the value of GHF/averaged hit rate is <91 for the 4 GHF which ranks the first four (Shen et al., 2020; Shapiro and Ritzwoller, 2004; Purucker, 2012; An et al., 2015), and >107 for the other 4 GHF which ranks the last four. We do not think it add more interesting information to the ranking.

Table S1. Warm hit rate, cold hit rate, averaged hit rate, GHF/averaged hit rate for the modelled results with eight GHFs in Table 2.

| GHF | Mean GHF (mW m$^{-2}$) | warm hit rate | cold hit rate | averaged hit rate | GHF/ averaged hit rate |
|---|---|---|---|---|---|
| Martos et al., 2017 | 65 | 0.9560 | 0.1648 | 0.56 | 120.37 |
| Purucker, 2012 | 51 | 0.5283 | 0.8201 | 0.67 | 80.95 |
| Shen et al., 2020 | 58 | 0.6588 | 0.6564 | 0.65 | 89.23 |
| An et al., 2015 | 51 | 0.4340 | 0.7652 | 0.60 | 83.61 |
| Shapiro and Ritzwoller, 2004 | 58 | 0.5975 | 0.7822 | 0.69 | 90.63 |
| Stål et al., 2021 | 63 | 0.8750 | 0.2405 | 0.56 | 107.14 |
| Lösing et al., 2021 | 64 | 0.9313 | 0.2216 | 0.58 | 108.62 |
| Haeger et al., 2022 | 59 | 0.9688 | 0.1458 | 0.56 | 114.29 |

For your comment to Line 415-417, i.e. "I did it as well in Van Liefferinge and Pattyn, 2013 but doing the mean basal temperature is not physical as the temperature cannot reach a higher value than the pressure melting point. Therefore, it is more correct to compare each result one by one (if you use the basal temperature).", we agree. But Eisen et al. (2020) did not show the modelled basal temperature using each GHF, and their result is not available to download. So we remove the sentence of Line 415-417 in the revision.

All the best,

Brice Van Liefferinge